# Oxidative Distress Induces Wnt/β-Catenin Pathway Modulation in Colorectal Cancer Cells: Perspectives on APC Retained Functions

**DOI:** 10.3390/cancers13236045

**Published:** 2021-11-30

**Authors:** Teresa Catalano, Emira D’Amico, Carmelo Moscatello, Maria Carmela Di Marcantonio, Alessio Ferrone, Giuseppina Bologna, Federico Selvaggi, Paola Lanuti, Roberto Cotellese, Maria Cristina Curia, Rossano Lattanzio, Gitana Maria Aceto

**Affiliations:** 1Department of Clinical and Experimental Medicine, University of Messina, Via Consolare Valeria, 98125 Messina, Italy; tcatalano@unime.it; 2Department of Medical, Oral and Biotechnological Sciences, University “G. d’Annunzio” Chieti-Pescara, Via dei Vestini 31, 66100 Chieti, Italy; emira.damico@unich.it (E.D.); carmelo.moscatello@unich.it (C.M.); fedeselvaggi@hotmail.com (F.S.); roberto.cotellese@unich.it (R.C.); mariacristina.curia@unich.it (M.C.C.); 3Department of Innovative Technologies in Medicine & Dentistry, University “G. d’Annunzio” Chieti-Pescara, Via dei Vestini 31, 66100 Chieti, Italy; dimarcantonio@unich.it (M.C.D.M.); rossano.lattanzio@unich.it (R.L.); 4Department of Medicine and Aging Sciences, University “G. d’Annunzio” Chieti-Pescara, Via dei Vestini 31, 66100 Chieti, Italy; alessio.ferrone@unich.it (A.F.); g.bologna@unich.it (G.B.); paola.lanuti@unich.it (P.L.); 5Center for Advanced Studies and Technology (C.A.S.T.), University “G. d’Annunzio” Chieti-Pescara, Via dei Vestini 31, 66100 Chieti, Italy; 6Unit of General Surgery, Ospedale Floraspe Renzetti, Lanciano, 66034 Chieti, Italy; 7Villa Serena Foundation for Research, Via Leonardo Petruzzi, 65013 Città Sant’Angelo, Italy

**Keywords:** colorectal cancer, oxidative stress, Wnt/β-Catenin, APC, gene expression, protein expression, microenvironment

## Abstract

**Simple Summary:**

The dysregulation of Wnt pathways is involved in colorectal carcinogenesis. H_2_O_2_ differentially regulates the Wnt/β-Catenin pathway in colorectal cancer (CRC), but the molecular mechanisms remain unclear. Cellular stress might also stimulate APC protein production retaining some functions in N-terminus and lead to cancer progression. The effect of oxidative distress on Wnt/β-catenin signaling in the light of APC functions is unknown. We exposed starved HCT116, SW480 and SW620 CRC cell lines to H_2_O_2_-induced short-term oxidative stress. This treatment promoted the activation of β-Catenin and increased cytoplasmic APC. H_2_O_2_ regulated gene expression, related to cellular phenotype and stimulated both Wnt/β-Catenin-dependent and -independent signaling. These findings suggest that oxidative distress may influence APC functions in Wnt signaling and open up new perspectives to develop personalized therapeutic approaches for CRC.

**Abstract:**

Colorectal cancer (CRC) is a multistep process that arises in the colic tissue microenvironment. Oxidative stress plays a role in mediating CRC cell survival and progression, as well as promoting resistance to therapies. CRC progression is associated with Wnt/β-Catenin signaling dysregulation and loss of proper APC functions. Cancer recurrence/relapse has been attributed to altered ROS levels, produced in a cancerous microenvironment. The effect of oxidative distress on Wnt/β-Catenin signaling in the light of APC functions is unclear. This study evaluated the effect of H_2_O_2_-induced short-term oxidative stress in HCT116, SW480 and SW620 cells with different phenotypes of APC and β-Catenin. The modulation and relationship of APC with characteristic molecules of Wnt/β-Catenin were assessed in gene and protein expression. Results indicated that CRC cells, even when deprived of growth factors, under acute oxidative distress conditions by H_2_O_2_ promote β-Catenin expression and modulate cytoplasmic APC protein. Furthermore, H_2_O_2_ induces differential gene expression depending on the cellular phenotype and leading to favor both Wnt/Catenin-dependent and -independent signaling. The exact mechanism by which oxidative distress can affect Wnt signaling functions will require further investigation to reveal new scenarios for the development of therapeutic approaches for CRC, in the light of the conserved functions of APC.

## 1. Introduction

Colorectal cancer (CRC) is the third most prevalent cancer worldwide, since 1.9 million new cases and approximately 935,000 deaths occurred in 2020 according to the World Health Organization [1]. The incidence and mortality rates of this tumor vary widely at the global level and increase with the progressive adoption of Western lifestyles [2]. CRC usually takes years to develop and complex interactions between genetic and environmental factors occur during this process [3,4]. Colorectal carcinogenesis goes through a multistep process, arising in the majority of cases from an adenoma that has the potential to evolve into carcinoma by the accumulation of additional somatic mutations and epigenetic alterations through interactions with the tissue microenvironment [5]. These events are primarily associated with Wingless/It (Wnt)/β-Catenin signaling dysregulation [6,7]. Wnt signals are transduced into cells through the canonical (β-catenin dependent) and non-canonical (β-catenin independent) pathways [8,9,10].

The aberrant regulation of the Wnt pathway on the pathogenesis and progression of about 100% of CRCs has long been recognized [11,12,13,14]. Changes in Wnt/β-Catenin signaling molecules, including adenomatous polyposis coli (APC) truncating mutations, cause the generation of constitutive nuclear complexes between T cell factor/lymphoid enhancer factor (TCF/LEF) and β-Catenin and lead to target gene deregulated transcription [4]. Activating mutations in the Wnt pathway represent a critical early event in CRC pathogenesis and are responsible for the inappropriate regulation of numerous target genes coding for oncogenic transcription factors [11,15]. Besides tumors, Wnt/β-Catenin signaling seems to be involved in local and systemic inflammation, leading to oxidative stress conditions through its non-canonical pathways [16,17,18,19]. Furthermore, oxidative stress itself may also regulate Wnt/β-Catenin signaling in a redox-dependent manner [20]. Oxidative eustress results from an imbalance between the levels of reactive oxygen species (ROS) and the activity of the cellular antioxidant mechanisms. However, oxidative distress can deregulate oncogenic signaling pathways that are involved in CRC carcinogenesis. ROS including free radicals such as superoxide anion (O^2−^), and non-radical molecules such as hydrogen peroxide (H_2_O_2_), can play harmful or beneficial roles depending on their concentrations [21]. A progressive and sustained oxidative distress can lead to tumor development and progression [22]. It could be assumed that stress phenomena may interfere in the modulation of canonical and non-canonical Wnt signals according to the specific characteristics of the tumor. In recent years there has been a growing interest in studying the role of oxidative distress in the onset of CRC, which could also interfere with cancer progression and therapeutic response [23,24,25,26]. H_2_O_2_ is also a signaling molecule involved in the regulation of cell proliferation and apoptosis [27,28,29]. Although, many studies have shown that in CRC cells the Wnt/β-Catenin pathway is differentially regulated by H_2_O_2_, the underlying molecular mechanisms are only partially known [30,31,32]. In CRC development, the inactivation of APC, by mutation or methylation, dysregulates Wnt/β-Catenin signaling. In addition to this, APC appears to contribute to tumorigenesis through other cellular processes [33]. Recently, new hypotheses have been formulated on the role of the gain of function of the truncated APC protein in addition to its loss of function [34]. It has also been proposed that certain stress conditions might promote the production of mutant APC proteins that retain some functions in their N-terminal portion [35,36]. However, the mechanisms that might define the modulation of Wnt/β-Catenin signaling and APC due to oxidative stress in CRC have not yet been investigated. In this study, our experimental goal was to induce momentary functional distress in CRC cells by mimicking a state of quiescence. To this aim, we initially tested the effect on the viability of H_2_O_2_ concentrations that could be used to induce short-term oxidative distress in CRC cells with a different β-Catenin and APC phenotype status. Therefore, we assessed the gene expression of APC and molecules characteristic of Wnt signaling β-Catenin-dependent or –independent. The analysis of correlation and modulation of APC gene expression allowed us to evaluate the conditions of acute oxidative stress at which we could observe a relationship with cell cycle, APC protein and other key components of Wnt/β-Catenin signaling. Our results showed that acute oxidative distress modulates canonical and non-canonical Wnt signals depending on tumor-specific characteristics and probably on APC protein status. For the first time, we demonstrated this effect in CRC cells with short-term exposure to external concentrations of H_2_O_2_, above estimated physiological ranges [27]. Specifically, H_2_O_2_induced gene expression of components of the canonical Wnt/β-Catenin pathway in CRC cells with mutated APC, whereas CRC cells with wild-type APC preferentially activated gene expression of the β-Catenin independent pathway. Moreover, the stress condition induced increased protein expression of β-Catenin and APC in primary tumor-derived cells both with or without APC mutation; in particular, APC showed a decrease in its full-length isoform and an increase in its stress-sensitive short isoform.

## 2. Materials and Methods

### 2.1. Cell Cultures and Treatments

Colon cancer cell lines, HCT116, SW480 and SW620 were obtained from American Type Culture Collection (ATCC, Manassas, VA, USA). HCT116 and SW620 were cultured at 37 °C in DMEM medium containing 10% fetal bovine serum (FBS), 100 U/mL penicillin/streptomycin and 2 mM L-glutamine (EuroClone, Pero, Italy). SW480 cell line was propagated in RPMI-1620 (EuroClone), supplemented with 10% FBS, 2 mM L-glutamine and 100 U/mL penicillin/streptomycin. For the experiments, cells at sub-confluence (70%), were starved using a medium with 0.2% FBS (serum-free medium) cells overnight in order to reduce basal cellular activity [37]. Then, the starved cells were acutely stimulated with H_2_O_2_ (Sigma-Aldrich, Milan, Italy) as an oxidizing agent, for the times and concentrations described below.

### 2.2. Cell Viability and Metabolic Assay

We used a colorimetric assay to test cell viability and metabolic activity. HCT116, SW480 and SW620 cell lines were seeded in a 96-well plate at a concentration of 1.0 × 10^4^ cells/well and then sub-confluent (70%), cultures were starved in 100 μL of a low serum medium (0.2%) overnight to reduce basal cellular activity [37]. The cells were exposed to an increasing concentration (from 0.05 mM to 10 mM) of H_2_O_2_ as an oxidizing agent at different times (3, 6 and 24 h). This was followed by incubation with 10 μL/well of 2-[2-methoxy-4-nitrophenyl]-3-[4-nitrophenyl]-5-[2,4-disulphophenyl]-2H-tetrazolium, monosodium salt (MTS) assay (Promega, Milan, Italy) at 37 °C for 1 h. The absorbance was measured at 490 nm with a microplate reader by GloMax-Multi Detection System (Promega). For each experimental condition, five replicates were performed in three independent experiments.

### 2.3. Gene Expression by Real-Time Quantitative PCR Analysis (qRT-PCR)

For gene expression experiments, the cells were seeded on a 10-cm plate at a density of 3.2 × 10^4^; at sub-confluence (70%), they were starved overnight using a medium with 0.2% FBS, then the cells were acutely stressed with 2 mM and 10 mM H_2_O_2_ for 30 min. Total RNA was isolated from HCT116, SW480 and SW620 cell lines untreated and treated with the oxidizing agent H_2_O_2_, using EuroGold TriFast™ (EuroClone, Pero, Italy), according to the manufacturer’s instructions. RNA samples were assessed for purity and quantified by Nanodrop 1000 Spectrophotometer (Applied Biosystems, Thermo Fisher Scientific, Waltham, MA, USA). The synthesis of complementary DNA (cDNA) was performed employing the GoTaq^®^ 2 Step RT-qPCR Kit (Promega) according to the manufacturer’s instructions. The mRNA levels were evaluated by a SYBR green quantitative real-time PCR (qRT-PCR) analysis using the StepOne™ 2.0 Real-Time PCR system (Applied Biosystems). Data were analyzed using the comparative Ct method and were graphically indicated as 2^−ΔΔCt^ ± SD. In accordance with the method, the mRNA amounts of the target genes were normalized by the ratio on the median value of the endogenous housekeeping, β-Glucuronidase (GUSB) gene, obtained in each treated cells vs. untreated (quiescent) cells. Targets and reference genes were amplified, at least in triplicate, in a volume of 10 μL containing 1μL of cDNA template, 0.2 μL of primers mixture and 5 μL of GoTaq^®^ 2-Step RT-qPCR system (Promega) according to the manufacturer’s instructions. The cycling conditions were performed as follows: 10 min at 95 °C and 40 cycles of 15 s at 95 °C, followed by 1 min at 60 °C and final elongation of 15 s at 95 °C. The sequences of paired oligonucleotides were:5′-GCTTGATAGCTACAAATGAGGACC-3′ and 5′-CCACAAAGTTCCACATGC-3′for *APC*; RefSeq: [NM_000038]5′-CCCATGCACCTGGTTCTACT-3′ and 5′-CCAAGCCACAGGGATACAGT-3′for *LRP-6*; RefSeq: [NM_002336]5′-ATGGGTTCTGCCAGCCTTAC-3′ and 5′-TAGACGTGCCGATCATGGTG-3′for *ROR*-2; RefSeq: [NM_004560]5′-CATGAACCGCCACAACAAC-3′ and 5′-TGGCACTTGCACTTGAGGT-3′for *WNT-3a*; RefSeq: [NM_033131]5′-CTCATGAACCTGCACAACAACG-3′ and 5′-CCAGCATGTCTTCAGGCTACAT-3′for *WNT-5a*; RefSeq: [NM_03392]5′-CCAACTTGCCATCAATGAATAA-3′ and 5′-GGCATCTGATTGGAGTGAGAA-3′for *BCL-9*; RefSeq: [NM_004326]5′-GAC GAG ATG ATC CCC TTC AA-3′ and 5′-AGG GCT CCT GAG AGG TTT GT-3′for *LEF-1*; RefSeq: [NM_016269]5′-TCGACATGGAGTCCCAGGA-3′ and 5-GGCGATTCTCTCCAGCTTCC-3′for JUN/AP-1; RefSeq: [NM_002228]5′-AGCCAGTTCCTCATCAATGG-3′ and 5′-GGTAGTGGCTGGTACGGAAA-3′for *GUSB*; RefSeq: [NM_000181]

### 2.4. Flow Cytometry and Cell Cycle Assay

The effect of limit stress on the cell cycle at short time points was evaluated by a fluorescence-activated cell sorting (FACS) analysis on tumor cell lines.

Flow cytometry cell cycle analyses were carried out as already reported [38,39]. The cells were seeded on a 6-cm plate at a density of 0.8 × 10^6^; at sub-confluence (70%), the cells were starved overnight (with 0.2% FBS), were treated with H_2_O_2_ 2 mM for short time (15 and 30 min) and then 5 × 10^5^ cells/sample were fixed by adding 500 μL of 70% cold ethanol and then stored at 4 °C. After at least 24 h, samples were washed and stained with 500 μL of a solution composed by 50 μg/mL of propidium iodide (PI, Sigma) and 200 μg/mL of RNAse (Sigma). Cells were incubated overnight at 4 °C in the dark and then acquired by flow cytometry.

PI fluorescence data were collected using linear amplification and at least 10,000 events were recorded for each sample. Samples were acquired on a FACSCanto II flow cytometer (Becton Dickinson Biosciences, Franklin Lakes, NJ, USA). Data were analyzed using the FlowJo software (v8.8.6, TreeStar, Ashland, OR, USA).

### 2.5. Wound Healing

To quantify the oxidative stress effect on starved CRC cell migration, we used a wound healing scratch assay on two-dimensional surfaces. The cells were cultured in triplicate on dishes in a 12-well plate at a density of 0.1 × 10^6^, to create a monolayer confluency, on which a straight line (an incision-like gap) was made with a pipette tip. To remove floating cells and debris, we rinsed the cells with PBS. Then, the medium was replaced with a starvation medium (0.2% FBS) overnight and then added with 0.05 mM H_2_O_2_. The scratch area was photographed immediately (T0) and after 24 h of incubation at 37 °C.

### 2.6. Preparing Cell Blocks from HCT116, SW480 and SW620 Cell Lines and Performing the Immunocytochemical (ICC) Stainings

For immunocytochemical experiments, the cells were seeded in the T75 flask at a density of 2.1 × 10^6^; at sub-confluence (70%), starved overnight using a medium with 0.2% FBS, then the cells were acutely stressed with 2 mM H_2_O_2_ for 30 min. For each tumor cell line, at least 8 × 10^6^ cells were prepared. The cells were trypsinized for 5 min at 37 °C and centrifuged at 1200 rpm for 5 min. Pellets were re-suspended in PBS and transferred in 2-mL tubes and then centrifuged two times at 1200 rpm for 5 min. Pellets were then fixed in 10% neutral buffered formalin (pH 7.2–7.4) for 60 min, centrifuged and placed in cassettes and embedded in paraffin using the Leica ASP 300 automatic tissue processor.

Immunocytochemistry (ICC) was carried out on 5-micrometer cell block sections stained with primary antibodies, as reported in Table 1. Antigen retrieval was performed by microwave treatment at 750 W (10 min) using the citrate buffer (pH 6.0) for the ICC staining with the anti-FZ-6, -APC, -β-Catenin and -E-cadherin antibodies, and the EDTA antigen retrieval buffer (1 mM EDTA, 0.05% Tween 20, pH 8.0) for the ICC staining with the anti-Cyclin D1 antibody. The anti-mouse and the anti-rabbit EnVision kits (Agilent, K4001 and K4003, respectively) were used for signal amplification, as appropriate. Furthermore, 3,3′-diaminobenzidine (DAB) was used as chromogen. Cell nuclei were counterstained with hematoxylin. In control sections, the specific primary antibodies were replaced with non-immune serum or isotype-matched immunoglobulins. The expression of molecular markers was assessed according to the presence of distinct specific staining in tumor cells.

### 2.7. Western Blotting

Total proteins were isolated from sub-confluent starved cells in 10-cm plates (3.2 × 10^4^), after 2 mM H_2_O_2_ stress for 30 min. The extraction was conducted using a RIPA lysis buffer (Cell Signaling Technology, Beverly, MA, USA). Protein concentrations were determined using the BCA protein assay (Thermo Fisher Scientific, Waltham, CA, USA). An equal amount of total proteins was separated on 4–20% SDS-PAGE pre-cast gel electrophoresis (Bio-Rad Laboratories, Hercules, CA, USA) and transferred onto PVDF membranes (GE Healthcare, Chicago, IL, USA). Then, after blocking, the membranes were incubated with the primary antibody overnight at 4 °C. The following primary antibodies were used: β-Catenin (Cell Signaling Technology) and Active-β-Catenin; APC (Merck-Millipore, Burlington, MA, USA); β-actin (Sigma-Aldrich, St. Louis, MI, USA) was used as a protein loading control. Secondary antibodies were HRP-conjugated anti-rabbit or anti-mouse (Bethyl Laboratories, Montgomery, TX, USA). The immune complexes were visualized using the ECL Western blot detection system (EuroClone) by using AllianceLD2 hardware (UVItec Limited, Cambridge, UK).

### 2.8. Statistical Analysis and Tools

All measurements were made after three independent experiments, and a representative value from all experiments plus standard deviation is shown for each data point. Results were subjected to a t-test or a one-way analysis of variance (ANOVA) as appropriate. All p values were two-sided, and a *p* value less than 0.05 was considered significant. All analyses were performed using IBM SPSS Statistics for Windows (version 20, IBM Corp., Armonk, N.Y., USA). The Multiexperiment Viewer program v4.9.0 (MeV v4.9.0, J. Craig Venter Institute, La Jolla, CA, USA) [40] by the average linkage hierarchical clustering with Pearson correlation was used to assess molecular clustering and correlations of gene expression modulation in response to H_2_O_2_.

## 3. Results

To evaluate the impact of the acute oxidative distress on the Wnt/β-Catenin pathway we used human CRC cell models, in the light of the APC retained functions.

We analyzed the effects induced by the exposure to H_2_O_2_ on viability and proliferation of HCT116, SW480 and SW620 cell lines. The choice to investigate CRC cell line models characterized by different behavior of the Wnt signaling was supported by their mutation pattern in Wnt pathway genes. The HCT116 cells were characterized by *APC* wild type, mutated β-Catenin (*CTNNB-1*), *LRP6* and *ROR2* [41] as well as MSI that was caused by biallelic mutation in MLH-1 gene [42]. The SW480 cell line and metastatic SW620 cells, derived from the same patient, were characterized by mutated APC and MSS [43].

The mutational status of Wnt pathway molecules in these cellular models is shown in Table 2.

### 3.1. Cell Viability after Oxidative Distress Induced by H_2_O_2_

In order to evaluate the effect of oxidative distress induced by H_2_O_2_ on cell viability and proliferation, HCT116, SW480 and SW620 cells were treated as described in Methods. An MTS analysis was performed to assess cell viability. The cells were exposed to H_2_O_2_ as an oxidizing agent for different times and concentrations (0.05, 0.5, 1, 2, 5, 10 mM for 3 h, 6 h, 24 h) (Figure 1). For each experiment, five replicated wells were assayed per clone. Cell viability values were calculated as means and compared to untreated vs. quiescent cells (Q). MTS assays revealed that H_2_O_2_ treatments induced different effects on the activation or inhibition of cell viability in time and dose-dependent manners in each cell line. A significant 70% rapid decrease (*p* < 0.001) of SW480 was observed at all H_2_O_2_ concentrations and persisted all times during the treatment (Figure 1a), in contrast to SW620 and HCT116 cell lines. In particular, the metastatic cells SW620 lost viability only after 24 h of stress (Figure 1b), while HCT116 after 3 h of treatment with H_2_O_2_ at a concentration of 2 mM showed a slight recovery of viability (Figure 1c).

### 3.2. Gene Expression Analysis by qPCR Real Time

The experiments were performed on starved HCT116, SW480 and SW620 cells exposed to H_2_O_2_ [2 mM] and H_2_O_2_ [10 mM] for 30 min to evoke an acute oxidative distress on quiescent cells.

H_2_O_2_ treated and not treated HCT116 cells revealed no change in *WNT3a* gene expression, while the treatment with 2 mM or 10 mM H_2_O_2_ upregulated the basal expression of non-canonical *WNT5a* and *ROR2* by H_2_O_2_ concentration-dependent hormetic effects. It should be noted that the HCT116 cell line carries the known pathogenic mutation *ROR2* R302H, c.905G>A (Table 2). After HCT116 cells’ exposure to 2 mM H_2_O_2_, gene expression profiling showed downregulation in *LRP6* and *BCL9*, whereas the treatment with a higher H_2_O_2_ concentration induced an increase in *LRP6* expression and a slight reduction of *BCL9* compared to untreated cells. Moreover, HCT116 cells harbor DNA mutation W419R, c.1255T>C in the canonical *LRP6* gene (Table 2). Exposure to H_2_O_2_ [10 mM] enhanced *LEF1* and significantly reduced *APC* expression while at low H_2_O_2_ concentration, *APC* resulted markedly upregulated and *JUN/AP1* slightly declined below the value observed in quiescent HCT116 cells (Figure 2).

In SW480 cells, acute stress induced by the treatment with H_2_O_2_ regulated gene expression of both Wnt pathways by a concentration-based modulation. In particular, H_2_O_2_ [2 mM] repressed the expression of Wnt3a and Wnt5a ligands. At this same concentration, H_2_O_2_ upregulated the expression of *LRP6* and *ROR2* co-receptors, as well as of *APC* and *JUN/AP1*, and slightly downregulated *BCL9* and *LEF1*. The treatment of SW480 cells with H_2_O_2_ [10 mM] led to a significant increase in the expression of LRP6, *ROR2*, *BCL9* and *JUN/AP1*, as well as to the downregulation of *WNT3a*, *WNT5a* and *APC*, and a little decrease in *LEF1* expression (Figure 2).

The effects of H_2_O_2_ treatment on the SW620 cell line resulted in substantial changes in gene expression of canonical and non-canonical Wnt pathways in a concentration-dependent way. H_2_O_2_ [2 mM] strongly increased co-expression of Wnt3a and Wnt5a ligands and slightly lowered *ROR2* expression, while *LRP6*, *APC*, *BCL9*, *LEF1* and *JUN/AP1* were reduced or completely abolished. A higher concentration of hydrogen peroxide upregulated *Wnt3a*, *ROR2* and *LEF1*, and abolished or strongly decreased the expression of all other genes (Figure 2).

### 3.3. Gene Expression Cluster Analysis

To assess the correlate effects of H_2_O_2_ on the Wnt pathway we compared the expression levels of Ligands: *WNT3a*, WNT5a; Coreceptors: *LRP6* and *ROR2*; β-Catenin controllers: *APC* and *BCL9* gene expression modulation after 30 min of oxidative stress induced by H_2_O_2_ 2 mM and 10 mM using the Multiexperiment Viewer v4.0 (MeV4.0) program [40] (Figure 3). Under acute oxidative stress conditions, HCT116 cells showed a close correlation between *APC* and *WNT3a* and then with *WNT5a* close to *ROR2* (co-receptor independent of β-Catenin), *BCL9* and *LEF1*. In these cells, the expression of the *LRP6* co-receptor (β-Catenin-dependent pathway) correlates with that of the transcription factor *JUN/AP1* (Figure 3a). The *APC* expression showed a strong relationship with *LRP6* in SW480 but also with *JUN/AP1* in its metastatic cells SW620 (Figure 3b,c).

### 3.4. Cell Migration under Oxidative Distress by H_2_O_2_ [0.05 mM]

The wound healing assay showed a different migratory profile in HCT116, SW480 and SW620 cell lines after the H_2_O_2_ low dose (0.05 mM for 24 h). Use of H_2_O_2_ induced HCT116 and SW620 cell migration and a greater closure of the scratch area already after 24 h of treatment, as compared with the respective untreated cells. In contrast, the closure of the gap area in the wound healing assay was not observed in SW480 cells (Figure 4).

### 3.5. Cell Cycle Analysis

Based on the MTS results, by flow cytometry we evaluated the acute oxidizing effect on cell cycle phase distribution in HCT116, SW480 and SW620 cells treated with 2 mM H_2_O_2_, after 15 and 30 min. The percentages of cells in G0/G1, S and G2/M phases of the cell cycle, as well as the apoptotic population, were calculated using FlowJo software v8.8.6. When compared to untreated starved cells, the FACS analysis in 2 mM H_2_O_2_-treated HCT116 cell line after 15 min showed little variations in DNA content that slightly increased in the G0/G1 phase as well as in the S phase and reduced in the G2/M phase (Figure 5a). After 30 min, cell percentage diminished in the G0/G1 and increased in the S and G2/M phases (Figure 5a). These data suggested that the H_2_O_2_ treatment of HCT116 did not substantially modify the G1 and G2 phases at short time points. Nevertheless, the comparison of the results after revealed an increase in the S and G2/M phases at 30 min vs. 15 min (Figure 5a). Interestingly, these results were in line with MTS assay data and suggested that the use of 2 mM H_2_O_2_ induced an HCT116 cell viability not associated with an enhance of cell cycle progression, but rather caused by a higher metabolic capacity of these cells than the untreated control cells (Figure 5a). We also evaluated the different effects of H_2_O_2_ treatment on cells derived from the primary tumor (SW480) and corresponding metastasis (SW620). In response to 2 mM H_2_O_2_ treatment, SW480 cells showed different distribution patterns in each cell cycle stage when compared to the untreated control (Figure 5b). After 15 min of exposure to the oxidizing agent, an increase was observed in the percentage of cells at the G0/G1, and a small raise was detected in the G2/M phase while the proportion of cells in the S phase decreased. After 30 min of incubation with the same stimulus, the G0/G1 percentage of SW480 cells continued to increase, whereas the rate of cells in the G2/M phase was lower than the unexposed control cells. At the same time, a slight increase in DNA content was identified in the S phase (Figure 5b). These findings support the evidence that the presence of H_2_O_2_ favored G0/G1 arrest in response to the stimulus in over half of the SW480 cells. The study of the cell cycle in the SW620 cell line showed a high percentage of cells accumulated in the S phase after 15 or 30 min of H_2_O_2_ treatment. We also observed a significant decrease of cells in G2/M fractions in both times, while in the G0/G1 phases their percentage had not been substantially modified by H_2_O_2_ treatment. These results indicated that oxidative stress activated the S phase in the metastatic cell line, while it induced G0/G1 phase cell cycle arrest in a high percentage of the same cells as previously observed in the SW480 cell line (Figure 5c).

### 3.6. Protein Expression Assay by Western Blotting

The effect induced by acute H_2_O_2_ stress in starved HCT116, SW480 and SW620 cells was also assessed on APC, β-Catenin and Cyclin D1 protein expression. Western blotting analyses revealed downregulation of the full-length APC protein (300 kDa band) in all cell lines after H_2_O_2_ (2 mM × 30′) treatment, when compared to the analogous untreated cells. This was more evident in the primary SW480 cell line than in metastatic SW620 cells, both harboring an APC mutation. In contrast, the APC low molecular weight (LMW) band (approximately 70kD), corresponding to the truncating APC isoform, was found to be slightly increased after H_2_O_2_ treatment in HCT116 cells and strongly upregulated in SW480 cell lines, while SW620 cells exhibited reduced LMW APC levels (Figure 6) (Appendix A: Immunoreactive bands from original western blot). On the other hand, we observed β-Catenin upregulation in β-Catenin-mutant HCT116 cells, as well as its reduced expression in the SW620 cell line in response to H_2_O_2_ treatment. In addition, Western blotting analyses showed no change in the β-Catenin levels of SW480 cells (Figure 6). The expression of the active form of β-Catenin appeared slightly increased in all three cell lines. Acute oxidative distress did not produce a change in the expression of cyclin D1, which was constantly expressed in HCT116 cells and slightly modulated in SW620 and SW480 cell lines (Figure 6).

### 3.7. Expression of Wnt Pathway Components in HCT116, SW480 and SW620 Cell Block Sections

In HCT116, SW620 and SW480 cells cultured under quiescent and proliferative conditions, we investigated the effects of 2 mM H_2_O_2_ treatment (30′) on the expression of Wnt/β-Catenin signaling components, including FZ-6 receptor and transducers (E-cadherin, β-Catenin, APC and cyclin D1).

In β-Catenin mutant HCT-116 cells cultured in the quiescent condition, H_2_O_2_ treatment induced cytoplasmic immunoreactivity for APC and FZ-6 when compared to the untreated cells (Figure 7a). Instead, E-cadherin expression levels, β-Catenin, and cyclin D1 remained unchanged. Under proliferative conditions, H_2_O_2_ treatment inhibited the cytoplasmic expression of FZD6 shown by the HCT116 untreated cells (Figure 7b). In contrast, expression levels of β-Catenin and cyclin D1 remained unchanged compared to untreated cells, whereas E-cadherin significantly reduced its localization on the plasma membrane. In APC mutant SW480 quiescent cells, H_2_O_2_ treatment induced E-cadherin expression at the cytoplasm level without membranous localization (Figure 7c). Furthermore, the H_2_O_2_ treatment induced β-Catenin and APC membranous localization, and the nuclear expression of cyclin D1 was also observed (Figure 7c). The membranous expression level of FZ-6 was unchanged in treated SW480 cells compared to the untreated cells. When the SW480 cells were cultured in a proliferative condition, the nuclear localization of β-Catenin was only observed in cells grown without H_2_O_2_, whereas the treated cells did not show any nuclear staining (Figure 7d). In contrast, there were no differences in the expression of FZ-6, β-Catenin, and APC between treated and untreated cells.

In metastatic SW620 cells cultured under homeostatic conditions, untreated cells did not show any immunoreactivity related to the investigated Wnt/β-Catenin signaling molecules. Otherwise, the cytoplasmic expression of E-cadherin, membranous expression of FZD6, membranous and nuclear expression of β-Catenin and nuclear expression of cyclin D1were only observed in H_2_O_2_ treated cells (Figure 7e). In proliferative SW620 cells, nuclear localization of β-Catenin and cytoplasmic localization of APC were only observed in treated cells, whereas the expression levels of FZ-6 and Cyclin D1 did not change upon H_2_O_2_ treatment (Figure 7f).

## 4. Discussion

CRC is considered the third most common diagnosed cancer among males and the second in females [1]. Current cancer therapies primarily include a surgical approach followed by chemotherapy and/or radiation therapy. However, in many patients the cancer recurs or does not respond to initial treatment. One of the causes of recurrence/relapse has been attributed to altered ROS levels produced in the microenvironment of cancer cells [48] that may play a role in the switch between dormancy and metastatic spread [49]. On the other hand, a large number of anticancer drugs aim to eliminate cancer cells and drug resistance by increasing ROS production [28]. Therefore, it is very important to explore the modulation of oxidative stress response of deregulated signaling pathways in CRC cells because this could promote new personalized therapeutic approaches to fight the disease [50].

Increased ROS levels may activate pro-survival and pro-proliferative pathways but also the metabolic adaptation of cancer cells to the tumor environment [51] as a progressive process [22]. Even Wnt/β-Catenin signaling can be regulated by oxidative stress in a redox-dependent manner [20], although the cellular context in which this may occur has not been clearly defined.

In order to investigate Wnt/β-Catenin modulation in an oxidative distress context, we used starved CRC cell models to circumvent the influence of serum-mediated growth factors on cellular metabolism [27,37]. Preliminarily, we assessed various H_2_O_2_ concentrations to induce a short-term distress in three cell lines derived from two CRC adult male patients with different Wnt signaling behaviors (see Table 2).

At the physiological level, a detailed understanding of the spatial and temporal pattern of H_2_O_2_ in cells and tissues is a major challenge. Multiple factors can alter concentrations in both the extracellular environment and subcellular compartments [27]. The intracellular physiological range of H_2_O_2_ probably extends between 1 nM and 10 nM up to about 100 nM [27]. The physiological concentration gradient is estimated to be 100-fold higher in the intracellular compartment compared with the extracellular one. However, this gradient can vary with cell type, compartment and intracellular enzymatic activities, such that the highest external concentrations can be up to 650-fold higher than internal concentrations [27]. In normal cells, adaptive responses to stress are thought to occur even at higher concentrations, for example, to induce certain biological activities such as inflammatory response, growth arrest and cell death. Elevated H_2_O_2_ level is a feature of the tumor microenvironment compared with normal tissues [52]. However, it is difficult to discern between defined amounts of H_2_O_2_ when it comes to the microenvironment of tumor cells that, by their very metabolic activities, contribute to the production of a changing oxidant environment to which they can adapt. Starved tumor cells are also thought to be similar to dormant tumor cells known to be more resistant to stressful environmental changes [53]. In our work, we aimed to evaluate the effect of increased H_2_O_2_ in starved cells and at short time intervals to better discriminate their effects on canonical and non-canonical Wnt/β-Catenin signaling in the immediate term. In light of the considerations on H_2_O_2_ already reported above, in the viability test we used a range of H_2_O_2_ concentrations in the culture medium, starting from 0.05 mM up to 10 mM. After these treatments, we observed a nearly constant behavior in the three starved cells. For this reason, we chose the intermediate concentration of 2 mM and the highest concentration of 10 mM to test the effects of H_2_O_2_ on gene expression.

Our gene expression results demonstrated that the treatment with H_2_O_2_ [10 mM] downregulated *APC* on all three cell lines, while exposure of the cells to H_2_O_2_ [2mM] induced *APC* gene in an intracellular molecular context-dependent manner (Figure 2). Under acute oxidative stress conditions, in HCT116 cells *APC* narrowly correlated with *Wnt3a* as well as with *Wnt5a*, which was close to the *ROR2* co-receptor independent of β-Catenin, and with *BCL9* and *LEF1*. In these cells, the reduced expression of the *LRP6* co-receptor in the β-Catenin pathway correlated with that of the *JUN/AP1* transcription factor (Figure 3a). Increased *APC* expression showed a strong relationship with the upregulation of *LRP6* in SW480, and with *JUN/AP1* in corresponding metastatic cells SW620 (Figure 3b, 3c). Furthermore, an increase in both cytoplasmic APC protein (Figure 7) and activated β-Catenin (Figure 6) was observed in SW480 cells. Previous findings demonstrated that Wnt signaling hyperactivation in the compartments of intestinal stem and adult cells is associated with the CRC recurrence, since a high expression of Wnt3a and Wnt5a ligands confers a more aggressive cell phenotype [54,55,56]. Wnt3a ligand preferentially triggers the β-Catenin dependent pathway, while Wnt5a stimulates the β-Catenin independent pathway [8,9] (see graphical abstract).

We also observed that cellular distress by H_2_O_2_ [2 mM] promoted the S phase of the cell cycle in the metastatic cell line SW620, in contrast to the behavior observed in SW480 cells (Figure 5b,c). It should be considered that the tumorigenic SW480 and SW620 cells originated from the same donor but had different metastatic potential [57].

The exposure of the HCT116 cell line to H_2_O_2_ [2 mM] did not substantially modify the G1 and G2 phases at short time points (Figure 5a). These results were in line with MTS assay data and suggested that the use of 2 mM H_2_O_2_ induced an HCT116 cell viability not associated with an enhancement of cell cycle progression, but rather caused by a higher mitochondrial metabolic capacity of these cells than the untreated control cells (Figure 1 and Figure 2). In light of this, a moderate exposure to H_2_O_2_ can cause calcium release from non-mitochondrial intracellular stores without inducing apoptosis or necrosis [58]. In order to test the consequences of mild oxidative distress on cell growth and migration capacity, we evaluated the effect of H_2_O_2_ [0.05 mM] for a longer time (24 h) on our cell models (Figure 4). The increased metabolism in HCT116 cells could be associated with their dynamics of movement and polarity mediated by Wnt/planar cell polarity (PCP) pathway activation, rather than being related to cell viability and proliferation evidenced by MTS. On the contrary, 0.05 mM H_2_O_2_ conferred an inhibitory effect on the migration of SW480 cells. The HCT116 cell line exhibited microsatellite instability (MSI) and bear hMLH1 mutation; this deficiency is known to strongly accelerate colon carcinogenesis when combined with mild inflammation [59]. In an inflammatory response, activation of epithelial and immune cells triggers H_2_O_2_ production [60].

Protein expression by Western blot showed that oxidative stress induces increased expression of activated β-Catenin and Cyclin D1 in SW480 and SW620 cells. We were surprised to find a short isoform of APC of about 70kD in starved cells. In a 2008 paper, Brocardo and colleagues had identified an isoform of APC that appeared to be induced by stress, but displayed a maintenance of function in the N-terminal portion involving it in the mitochondrial response to stress [35].

Our results showed that oxidative stress exposure reduced full-length APC levels but increased the expression of the shorter isoform. Further studies will be required to assess the APC function in distress condition. However, the ICC data (Figure 7a,c,f) confirmed that in an oxidative stress microenvironment, the APC protein can be induced in cells with wild type APC but also in APC mutant cells such as SW480 and SW620. Notably, APC expression appeared more intense in HCT116 and SW480 starved primary tumor cells than in the proliferating ones.

## 5. Conclusions

From recent research, metabolically generated H_2_O_2_ has emerged as a central hub in redox signaling and oxidative stress. Malignant cancer cells are known to contain and tolerate higher ROS levels than normal cells, which is due to distorted cellular metabolism. Based on this feature, there is a growing interest in manipulating ROS levels in anticancer therapies to selectively kill cancer cells without affecting normal cells. Furthermore, some drugs also induce ROS and redox stress as secondary effects, and for some cancer chemotherapies, such effects lead to immunogenic cell death that helps in activating the immune system against the tumor cells [48].

The data presented in this study indicated for the first time that CRC cells, even when deprived of growth factors under acute oxidative distress conditions by H_2_O_2_, promote β-Catenin expression and modulate APC protein. In addition, H_2_O_2_ induces a differential gene expression in the analyzed models, depending on the tumor phenotype and leading to favoring both WNT/β–Catenin-dependent and -independent signaling.

However, the exact mechanism by which oxidative distress may affect the Wnt signaling function will require further investigations to reveal new scenarios needed in the development of CRC therapeutic approaches in the light of APC retained functions. Moreover, numerous proteins involved in key cellular processes exhibit the bimodality of the function in cancer, both acting as tumor suppressors and oncoproteins. These capacities may be determined by cellular as well as environmental contexts [61]. Clarifying the mechanism(s) underlying this context-dependent duality is essential to provide new insights into the personalized therapeutic strategies.

## Figures and Tables

**Figure 1 cancers-13-06045-f001:**
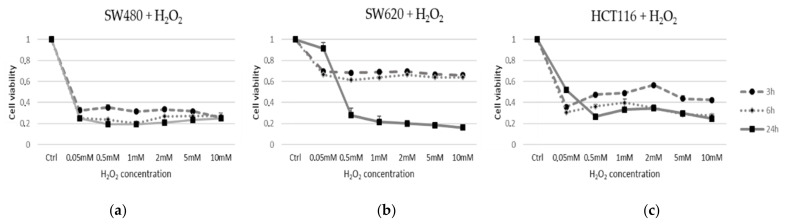
Cell line viability under acute oxidative distress condition. Cell viability was assayed using MTS Dye. (**a**) SW480 (MSS), (**b**) SW620 (MSS) and (**c**) HCT116 (MSI) cell lines were treated with H_2_O_2_ at concentrations of 0.05; 0.5; 1; 2; 5; 10 mM and timing of 3 h, 6 h, 24 h. Cell viability values were calculated as means and compared to untreated starved cells (Ctrl).

**Figure 2 cancers-13-06045-f002:**
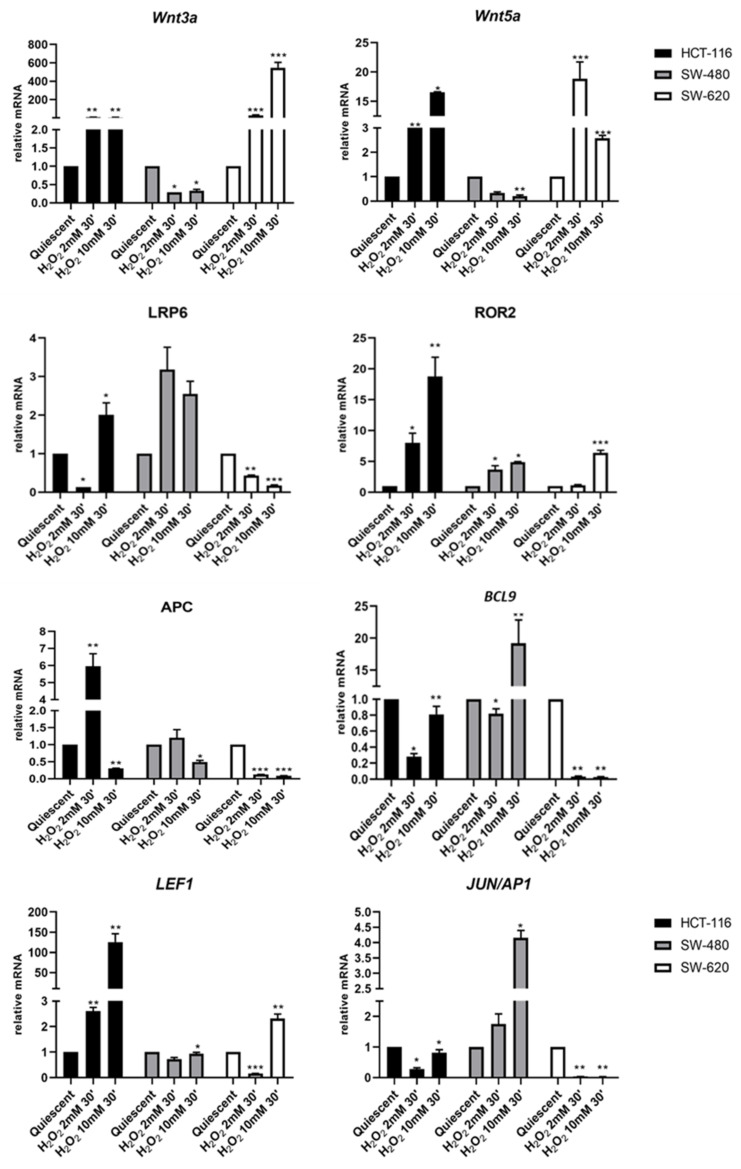
Expression of genes belonging to canonical and non-canonical Wnt signaling. Ligands: *WNT3a*, *WNT5a*; Coreceptors: *LRP6* and *ROR2*; β-Catenin controllers: *APC* and *BCL9* gene expression modulation after 30 min of oxidative stress induced by H_2_O_2_ 2 mM and 10 mM in HCT116, SW480 and its metastatic cells SW620. Gene expression was analyzed by real Time-qPCR. The histogram represented normalized data with *GUSB* gene. The results showed the average of three independent experiments. * *p* < 0.05, ** *p* < 0.01, *** *p* < 0.001 treated vs. quiescent cells.

**Figure 3 cancers-13-06045-f003:**
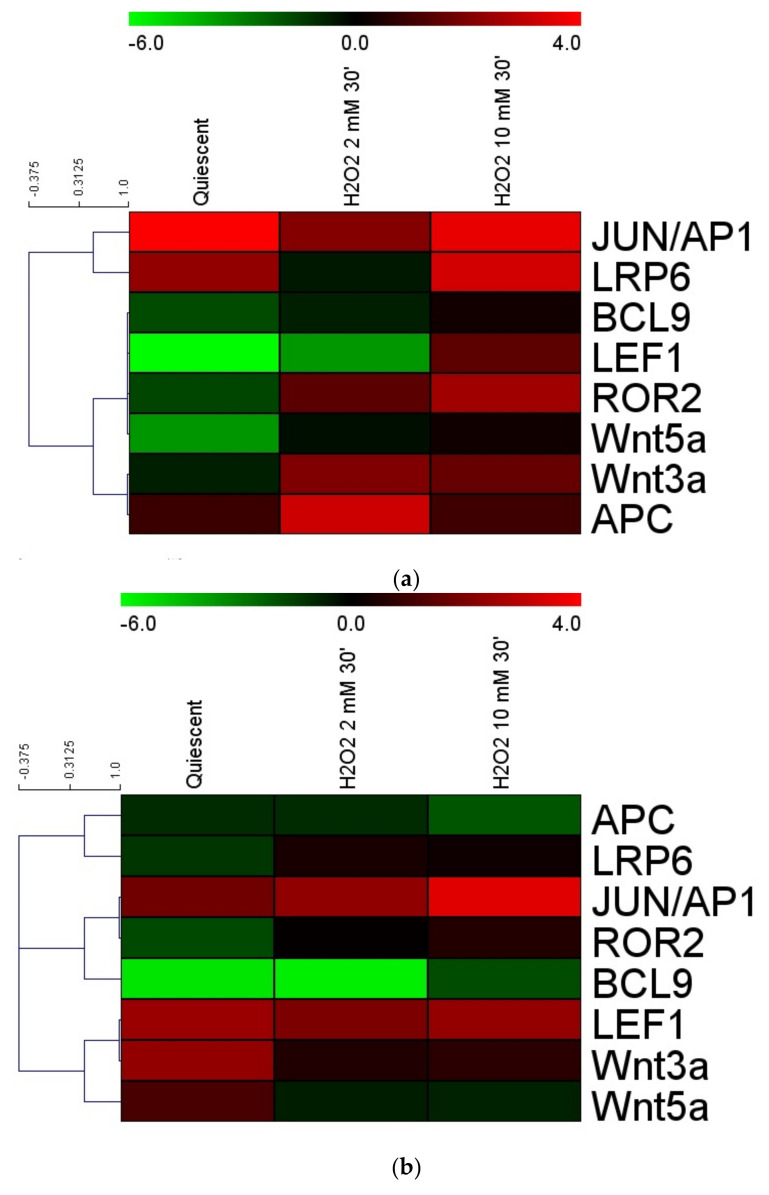
Gene expression cluster analysis by MeV4.9.0 in CRC quiescent cells treated by H_2_O_2_ [2 mM]: (**a**) HCT116 cells; (**b**) SW480 cells; (**c**) SW620 cells. The average linkage hierarchical clustering with Pearson correlation was used. The color scale at the top represents the log2 of every single gene expression value compared to housekeeping value ranging from. −11 (green) to 4 (red). The trees presented here are the neighbor-joining trees based on gene expression variation in response to different treatments.

**Figure 4 cancers-13-06045-f004:**
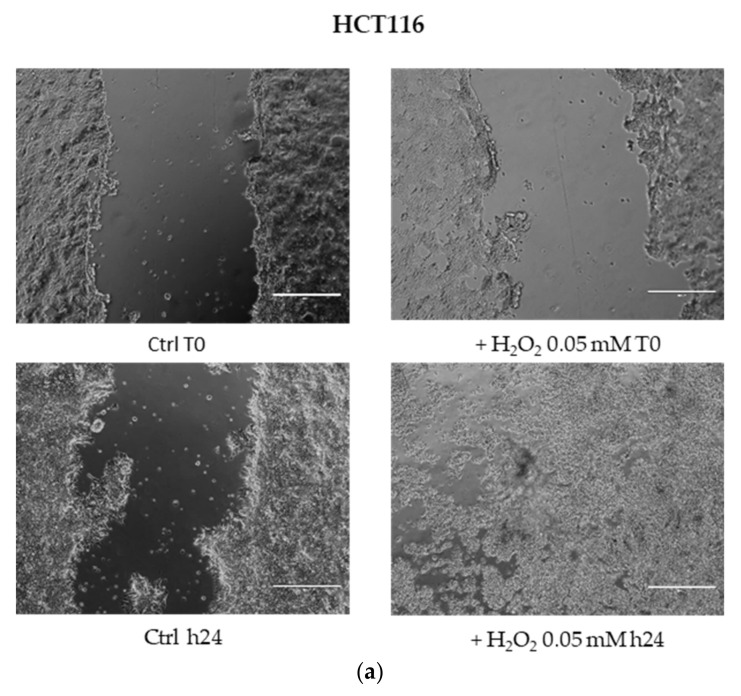
Wound healing assay. Cell migration was determined by in vitro closure of scratch area after 24 h 0,05 mM H_2_O_2_ treatment in the absence of serum: (**a**) HCT116 cells, (**b**) SW480 cells, (**c**) SW620 cells. Images were captured by a camera coupled to the EVOSTM XL Core Imaging System microscope (Thermo Fisher), with 10 × magnification, before (time-0) and after 24 h of treatment, and compared to control without H_2_O_2_.

**Figure 5 cancers-13-06045-f005:**
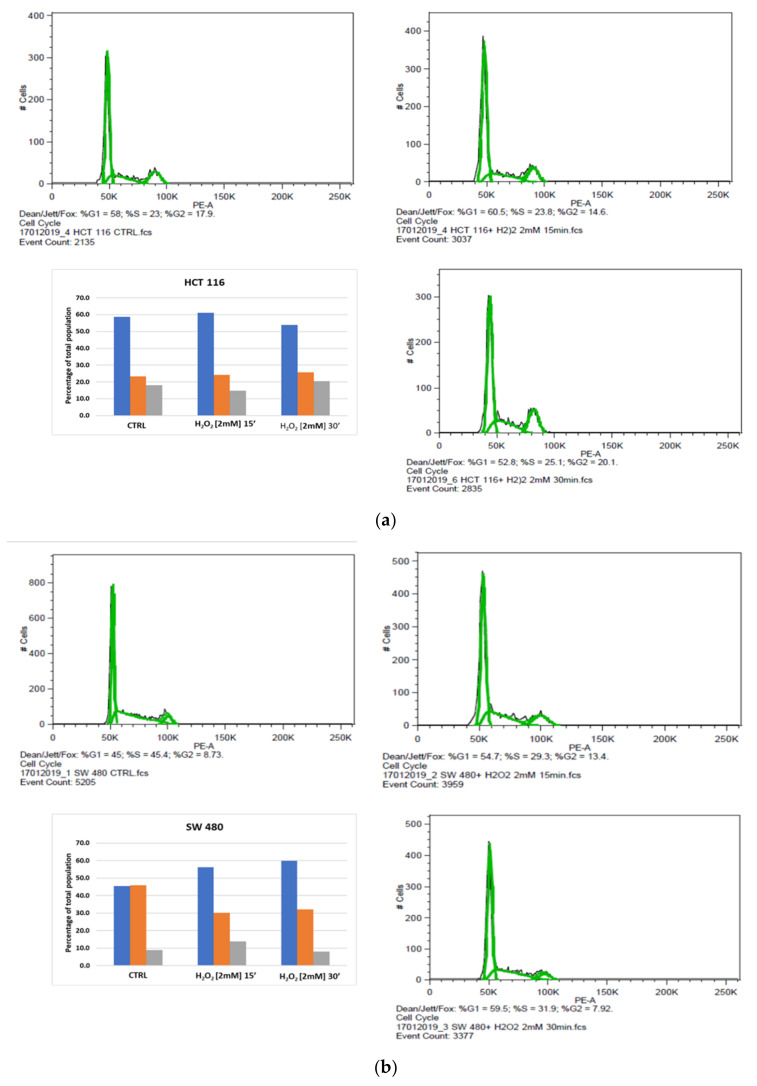
Effect of acute oxidative distress on the cell cycle of starved colon cancer cell lines. Profile analysis and relative percentages in cell cycle stages: (**a**) HCT116, (**b**) SW480, (**c**) SW620. Representative plots of FACS analysis of HCT116, SW480 and SW620 cells, untreated and 15 or 30 min after treatment with 2 mM H_2_O_2_. Graphics indicate percentages of cells in each phase of the cell cycle. Blue bars represent G1/G0 phase, orange bars represent S phase and gray bars represent G2/M phase.

**Figure 6 cancers-13-06045-f006:**
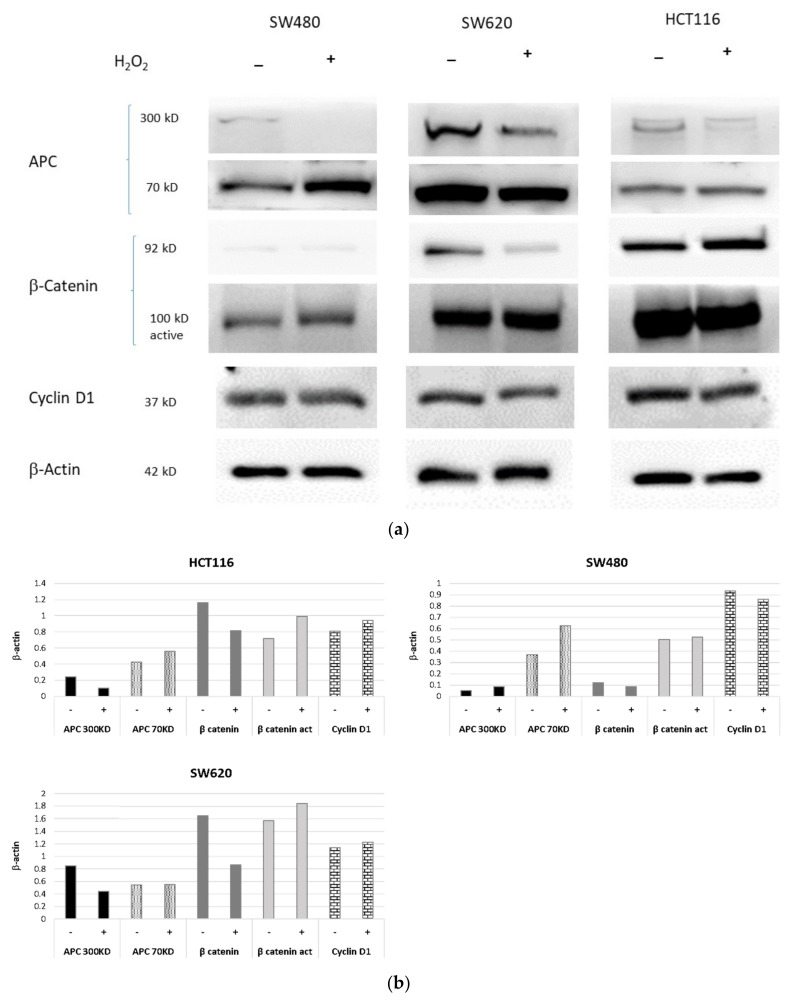
Protein expression of APC, β-Catenin and Cyclin D1 in HCT116 vs. SW480 vs. SW620 cells. The cells were starved overnight and treated with H_2_O_2_ [2 mM]. (**a**) Protein expression detected by Western blotting analysis are representative of three independent experiments. (**b**) The average expression levels of panel were determined by densitometric analysis and calculated in relation to the β-Actin level. kD: Kilodalton as protein molecular weight unit.

**Figure 7 cancers-13-06045-f007:**
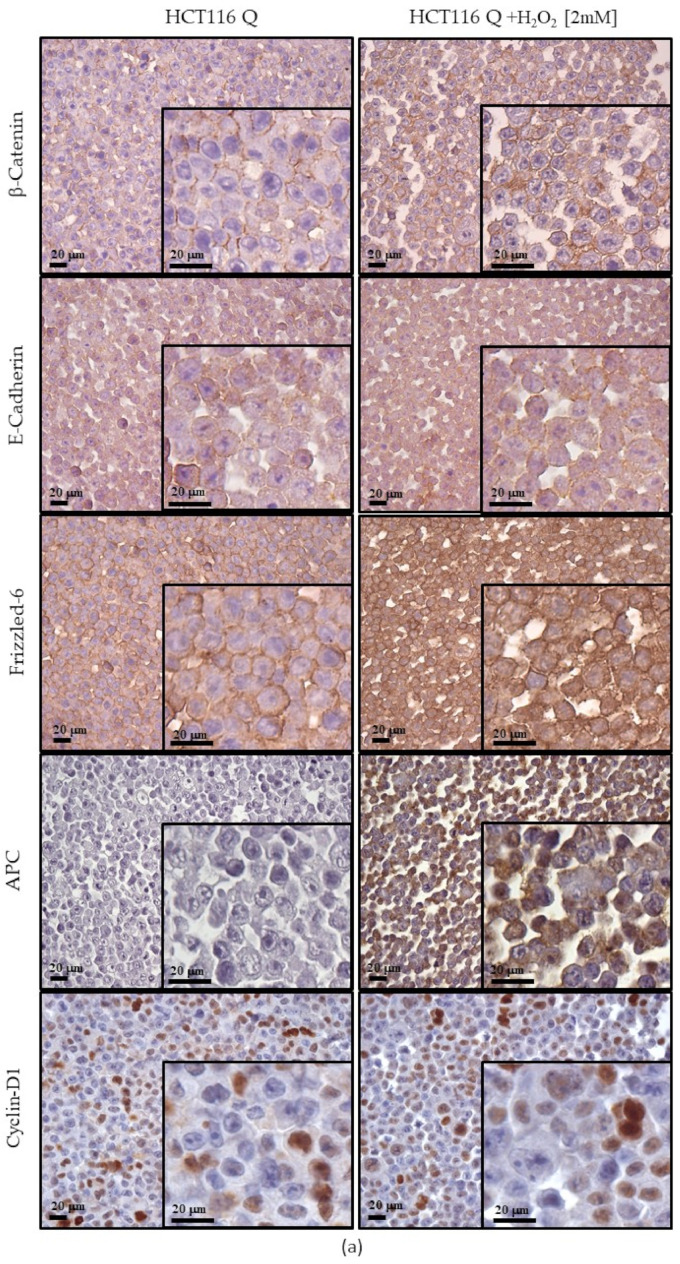
Expression of Wnt pathway components observed by ICC of cell block sections. In quiescent conditions HCT116 (**a**), SW480 (**c**) and SW620 (**e**) cells, in proliferative conditions HCT116 (**b**), SW480 (**d**) and SW620 (**f**) cells. Scale bars = 20 μm.

**Table 1 cancers-13-06045-t001:** List of antibodies.

Antibody	Company/Catalog No.	Type (Clone)	Dilution (Incubation)
FZ-6	Novus Biological/#NBP1-89702	Rabbit polyclonal	1:100 (ON)
APC	Thermo Fisher/PA530580	Rabbit polyclonal	1:100 (30′)
β-Catenin	BD Bio./610154	Mouse monoclonal (Clone 14)	1:3000 (60′)
E-cadherin	BD Bio./610181-82	Mouse monoclonal (Clone 36)	1:50 (30′)
Cyclin D1	Ylem/MCP511	Mouse monoclonal (P2D11F11)	1:25 (60′)

**Table 2 cancers-13-06045-t002:** Cell line characteristics and WNT mutation status.

Cells	MS	CIMP	CIN	*WNT3a*	*WNT5a*	*LRP6*	*WIF1*	*ROR2*	*APC*	*CTNNB1*	*LEF1*	*cJUN/AP1*
HCT116	MSI	+	-	wt	wt	W419R	met	R302H	wt	S45del	wt	Wt
SW480	MSS	-	+	wt	wt	wt	met	wt	Q1338 *	wt	wt	Wt
SW620	MSS	-	+	wt	wt	wt	met	wt	Q1338 *	wt	wt	Wt

MSI, microsatellite instability; MSS, microsatellite stability; CIMP, CpG Island methylator phenotype; CIN, chromosomal instability; wt, wild type, * SW480 and SW620 cell lines derived from the same patient [41,44,45,46,47].

## Data Availability

The datasets used and/or analyzed during the current study are available from the corresponding author upon reasonable request.

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
