# Peer review of "Oxidative Distress Induces Wnt/β-Catenin Pathway Modulation in Colorectal Cancer Cells: Perspectives on APC Retained Functions"

_cancers, 2021, doi:10.3390/cancers13236045_

Round 1

Reviewer 1 Report

Oxidative stress is involved in the pathogenesis and development of gastrointestinal disorders such as colorectal cancer (CRC). The authors have identified an important question “How is oxidative stress regulating the wnt-beta catenin signaling pathway in colorectal cancer cell models?” To answer this question, they chose to use HCT116 and SW480 and SW620 cell lines with CTNNB1 and APC mutations and H2O2 treatment to induce oxidative stress. Under these conditions the authors evaluated the gene expression patterns corresponding to both wnt canonical and non-canonical pathways.

comments:

  • It is not clear why the authors have specifically chosen the concentrations 2mM and 10 mM of H2O2treatment for their gene expression analysis experiments. For example, in the MTS assay for cell viability 0.5mM and 10 mM treatments did not show any significant differences in HCT116 and SW480 cells. How do we know the optimum H2O2 concentration that can model the redox environment with respect to the reactive oxygen species generated in the tumor microenvironment?
  • The western blot data in figure.6 requires to be repeated and reproduced to make any conclusions from the data. Also, why is there a truncated or low molecular weight APC band in HCT116 cells that have wild type APC?
  • Figure 5 can be further improved for publication quality. For example: by making the labels more visible, title of the bar graph has a type (cell cycle). The bar graph should have a labeled Y-axis. Also presenting error bars for these bar graph is advisable.
  • The manuscript needs to be thoroughly proof-read. For example, in line 129 (“For the experiment” is repeated), in line 227 (acutlely stressed with 2mM “H2O2” is missing), line 531(close to).

Author Response

We thank the reviewer for his careful reading of the manuscript and his constructive remarks. We have taken the comments on board to improve and clarify the manuscript. Please find below a detailed point-by-point response to all comments (reviewers’ comments in black, our replies in blue).

Please see the attachment for details.

Reviewer 2 Report

The manuscript represents a comprehensive evaluation of the acute effects of H2O2 exposure to components of the Wnt/-catenin pathway in colon cancer cell lines. Results show the ability of H2O2 to modulate the activation of the canonical or non-canonical Wnt/-catenin pathways, depending on APC protein status (mutated or wild type). The results have potential implications for anti-tumor therapies based on a better understanding of the modulating role of ROS on colon carcinogenetic processes.

I have no major concerns regarding this work. My congratulations to the authors for conducting an such a comprehensive analysis and preparing this insightful paper.

Please find below some minor points to consider:

1.- Spell out APC on first use.

2.- Line 129: "For the experiments" is repeated.

3.- Lines 214, 216: Please substitute "wound healing" for "would healing".

4.- Line 271: You meant: "oxidative distress"? The word is missing.

5.- Line 515: "To eliminate the disease". It might seem too ambitious yet to consider that elimination of cancer is close based on antioxidant/ROS modulation therapies. I would suggest replacing the verb with 'to fight' or 'to help to control" the disease.

6.- "Under of". The preposition ('of') should be suppressed.

Author Response

Review Report (Round 1)

The manuscript represents a comprehensive evaluation of the acute effects of H2O2 exposure to components of the Wnt/b-catenin pathway in colon cancer cell lines. Results show the ability of H2O2 to modulate the activation of the canonical or non-canonical Wnt/-catenin pathways, depending on APC protein status (mutated or wild type). The results have potential implications for anti-tumor therapies based on a better understanding of the modulating role of ROS on colon carcinogenetic processes.

I have no major concerns regarding this work. My congratulations to the authors for conducting an such a comprehensive analysis and preparing this insightful paper.

Please find below some minor points to consider:

1.- Spell out APC on first use.

2.- Line 129: "For the experiments" is repeated.

3.- Lines 214, 216: Please substitute "wound healing" for "would healing".

4.- Line 271: You meant: "oxidative distress"? The word is missing.

5.- Line 515: "To eliminate the disease". It might seem too ambitious yet to consider that elimination of cancer is close based on antioxidant/ROS modulation therapies. I would suggest replacing the verb with 'to fight' or 'to help to control" the disease.

6.- "Under of". The preposition ('of') should be suppressed.

AUTORS RESPONSES

We thank the reviewer for careful reading of the manuscript and for appreciating our research.

We considered the reviewer's comments to improve and correct the manuscript.

We carefully checked the whole text to make the necessary corrections (shown in the text in blue color) and properly comprehend the manuscript.

Line 30: we spell out the acronym “adenomatous polyposis (APC)”

Line 75: we spell out Adenomatous Polyposis Coli (APC) in place of APC on first use

Line 129: we deleted the repeated words “For the experiments”

Lines 214, 216: we substituted “wound healing” for “would healing”

Line 515: we replaced the verb “to eliminate” with “to fight”

Line 587: we suppressed the preposition “of” in “under of”

Line 531: we substituted “closed to” with “close to”

Line 205: we substituted “stained by” with “stained with”

Line 251: we added full stop between the words “(Cell Signaling Technology)” and “Protein”

Line 287: we substituted “chromosomal” for “cromosomal”

Round 2

Reviewer 1 Report

Thank you for addressing the comments appropriately.